# Individual, Interpersonal, and Organizational Factors Affecting Physical Activity of School Adolescents in Pakistan

**DOI:** 10.3390/ijerph18137011

**Published:** 2021-06-30

**Authors:** Tayyaba Kiyani, Sumaira Kayani, Saima Kayani, Iffat Batool, Si Qi, Michele Biasutti

**Affiliations:** 1Department of Sport and Exercise Science, Zijingang Campus, Zhejiang University, Hangzhou 310058, China; kiyani@zju.edu.cn; 2Department of Psychology, College of Education, Zhejiang Normal University, Jinhua 321004, China; sumaira@zjnu.edu.cn; 3Department of Education, Pir Mehr Ali Shah Arid Agriculture University Rawalpindi, Punjab 46300, Pakistan; 4Department of Education, Women University of Bagh, Bagh 12500, Pakistan; saimakayani22@gmail.com; 5Faculty of Social Sciences, Arts and Humanities, Superior University, Lahore 54600, Pakistan; iffatbt@yahoo.co.in; 6Department of Philosophy, Sociology, Education and Applied Psychology (FISPPA), University of Padova, 35139 Padova, Italy

**Keywords:** physical activity, adolescents, individual level, interpersonal level, organizational level, social ecological model

## Abstract

The purpose of this study was to explore individual, interpersonal, and organizational factors that may influence the physical activity of adolescents (ages 10–14) in Pakistani schools. A set of questionnaires that included individual, interpersonal, and organizational factors and PA behavior was completed by the 618 students selected from Pakistani schools. Stepwise forward regression model was applied to check the possible effects of multilevel variables on physical activity and to extract the stronger predictors. The results showed that physical activity was significantly predicted by individual level factors such as self-efficacy, motivation, and attitude. Among the demographic correlates, gender, age, and BMI did not affect physical activity, while socioeconomic status and geographic characteristics had a meaningful association with PA. At the interpersonal level, adolescents’ perception of family support had a potential influence on physical activity, while there was no impact of friends/peers and teachers support on adolescents’ PA. A school environmental characteristic, such as PA facility, was positively related to PA; however, the impact of PA equipment, safety, and policy and PA culture were statistically non-significant. The findings suggest that public health intervention strategies aimed at promoting PA in adolescents should recognize multiple levels of influences that may either enhance or impede the likelihood of PA among adolescents.

## 1. Introduction

Physical activity (PA) is an essential element of physical and mental health [1,2,3]. However, a large number of school adolescents are not performing the sufficient activity in Pakistan [4,5]. According to Stepwise Approach to Chronic Disease Risk Surveillance (STEPS) survey in Pakistan, the inactivity among school adolescents is 41.5% [6]. The inactivity is not affecting the eastern countries [7] but adolescents around the world are also not meeting the recommended guidelines for PA [8]. Considering the importance of the development of PA behavior in later ages, it is necessary to inculcate such habits in early age [9].

Factors affecting PA are quite complex with a wide range of variability [10,11,12]. According to an ecological model of health promotion, factors influence PA at the individual level, interpersonal level, organization level, community level, and policy level [11,12,13,14]. This study is based on three levels of the model, namely the individual, interpersonal, and organizational levels. The individual factors can be gender, age, body mass index (BMI), geographic characteristic (GC), socioeconomic status (SES), attitude, motivation, and self-efficacy [15,16,17,18]. Interpersonal factors may be the support provided by parents, peers, or teachers [18,19,20], while organizational factors include school PA environment and policies [12,20,21].

Various previous studies have found demographic and psychological variables as individual factors associated with PA. Literature shows a decrease in PA with age, with a decline more apparent in girls than boys particularly in the period of adolescence [1,7]. Within Pakistan, gender difference was observed in sport participation [22], with male students more active than females [4]. However, there was no such trend in other studies [2,23]. Further, differences among different geographic groups, such as from rural and urban areas, of different cultures were detected [24,25,26]. For example, a Chinese study indicated that adolescents in urban schools are more active than adolescents in rural schools [24]. Moreover, a study from France (*n* = 2523, age = 14–18) indicated that socioeconomic status was associated with levels of PA and sedentary behavior for both boys and girls [27]. Research also highlighted that low socioeconomic status provides more opportunities for sedentary behavior and less PA [28], and high socioeconomic status offers more doors for engaging in activity and decreasing sedentary time [29]. Conversely, socioeconomic status is not relevant for PA of adolescents in school, according to previous research [12]. Although the associations between PA and BMI were mostly non-significant or largely inconsistent in some studies [23], few studies have found BMI an effective predictor for PA [10].

The three important psychological factors that were identified in all research are self-efficacy, motivation, and attitude. Several researchers found direct association of self-efficacy with adolescents’ PA in their review and suggested exploration of the construct further through intervention [15,16]. For instance, self-efficacy was one of the effective predictors of PA in a cross-sectional study for determining the multilevel factors affecting in-school leisure time PA of adolescents [18]. Also, motivation for PA is considered an important element to participate in sports [30]. It was pointed out that exercise self-efficacy and exercise motivation directly affects the PA of school adolescents [31]. Besides, D’Haese and his colleagues revealed a more positive attitude was related to an increase in average daily steps and sports during leisure [32]. For example, an intervention study exhibited a significant improvement in the exercise attitude among an experimental group in school adolescents [16]. Hence, improving students’ exercise self-efficacy, motivation, and attitude helps to promote students’ PA levels. However, the study on these factors in Pakistan needs attention.

Additionally, interpersonal variables, such as social indicators, have repeatedly been reported to affect PA behavior. Previous studies reveal that parental support for PA is positively linked with promotion of PA [33]. Support from parents include motivating adolescents for PA, serving as guardians for their sport activities, engaging adolescents in several activities, exercising with them, and spending time in teaching them to do physical activity or to play any sport [33]. Moreover, adolescents who have the support of peers or friends are more active physically [34]. For instance, a study with 2361 adolescents showed a strong relationship between physical activities of adolescents and those of their friends [35]. Besides, teachers’ support for sport participation was positively related to promotion in PA behaviors [36,37,38]. Conversely, a meta-analysis suggested that interpersonal support is not a strong predictor for PA of adolescent girls [39]. Therefore, additional research is needed to investigate and to clarify the importance of interpersonal support for providing basis for future interventions.

In addition to individual and interpersonal factors, organizational factors, such as availability of PA equipment, facilities, safety, school policy, and culture, are also important to increase PA of adolescents [40]. For instance, Zhang and Si [41] found that availability, accessibility, greening of schools, sport venues and recognition, implementation of policies, and sport culture and atmosphere were significantly related to PA of school students. Past studies have explored that high PA is the result when adolescents are exposed to recreational or sport facilities or equipment, such as school play grounds, parks, and gymnasiums [42]. A recent western study revealed that adolescents (*n*= 435, age range 9–14) having more access to exercise equipment, like parks containing sport fields, showed higher levels of PA [43]. Several review studies indicated that organizational factors, like school size, play areas, facilities, accessibility, equipment, and policies for PA in school, play significant role to promote PA of adolescents at school [41,44]. Moreover, researchers have found school differences among adolescents’ interaction with environmental factors [42], demonstrating that the organizational factors affecting PA are inconsistent [45,46] because some research exhibited individual or personal factors more influencing for the phenomenon [47]. Hence, more investigation based on organizational factors are suggested [7].

It is argued that organizational and interpersonal factors affect PA in addition to the individual factors. However, it is necessary to conduct more research to provide further evidence. Therefore, the major aim of this study is to identify the potential contribution of social ecological factors to PA for adolescents (age = 10–14 years) in Pakistan. It is hypothesized that individual variables would significantly affect PA of adolescents in Pakistan (H1). The interpersonal factors are expected to be directly related to PA of school adolescents in Pakistan (H2). It is further predicted that organizational factors are positively associated with PA of school adolescents in Pakistan (H3). To the best of our knowledge, no study on multilevel correlates of PA has been done in a south Asian country like Pakistan. We have selected only three levels of the social ecological model for the current study because it was difficult to extend our study to policy and community level factors due to the unavailability of resources. In addition, there is no research based on any of the two levels of the social ecological model together in Pakistan. A better understanding of the effects of the individual, interpersonal, and organizational factors may improve the design and characteristics of PA interventions to reverse declining PA levels and consequently improve the overall health of Pakistani adolescents.

## 2. Method

### Participants and Procedure

The study sample was conveniently taken from the four schools in Rawalpindi city of Pakistan. The population of the study is composed of all the students enrolled in these four schools. Participants were invited to participate on the basis of informed consent. After the research proposal was approved from institutional review board of Zhejiang University, data were collected during October–November 2020, when schools were re-opened during COVID-19. All formal permissions were taken from the local authorities in Rawalpindi city. Data collection involved the collaboration of a local researcher. Seven hundred questionnaires were distributed among students in different schools. Of these, 664 questionnaires were returned, resulting in a response rate of 94.86%. The survey was not affected by COVID-19, as the government allowed the re-opening of schools, thus ensuring the students’ attendance. The results may be different from the normal conditions. Data were checked for missing values, and 28 cases were removed. Then, the researcher screened data and conducted all analyses in IBM SPSS v.20 (IBM, Armonk, NY, USA). Mahalanobis distance was calculated, and 8 outliers were found, and the cases were deleted. Finally, the sample consisted of 618 school adolescents (51.5% males, 48.5% females) with an age range of 11 to 14 years (SD = 0.97). Socioeconomic status was measured in terms of social classes based on 3-point scale containing lower class, middle class, and upper middle class. A total of 48.7% of the students were from lower-middle class, 50.2% were from middle class, and 1.1% were from upper-middle class. The students from rural areas were 49.7%, while those from urban areas were 50.3%.

## 3. Measures

### 3.1. Demographic Factors

Demographic characteristics of the students included demographics (5 items, i.e., age, gender, grade, weight, height), GC (1 item responded as selecting from two areas: 1 = rural and 2 = urban), and socioeconomic status (1 item, responded as selecting from social classes: 1 = lower-middle class, 2 = middle class, and 3 = upper-middle class). BMI was measured through self-reported weight and height by means of the formula: BMI = weight (kg)/height^2^ (m) [48].

### 3.2. Intrapersonal Factors-Related Measurements

(1)Self-efficacy. The exercise self-efficacy scale by [49] was adopted in this study. The scale is part of the self-efficacy scale for school adolescents, containing 9 items. The scale was used in its original English version, as the official and institutional language in Pakistan is English.(2)Internal and external motivation. Internal and external motivations for PA were measured with scales developed in a previous study by [50] based on SDT [51,52,53]. The internal motivation scale consists of three items. An example item is, “I enjoy it”. The external motivation scale also contains three items. The example item is, “My parents, other family members, or friends tell me to do it”. All items are measured with options 1 = not at all true to 5 = very true.(3)Attitude. Attitude toward exercise was measured using 5-point bipolar adjective scales developed by [54]. We assessed both instrumental (useful–useless, harmful–beneficial, wise–foolish, bad–good) and affective (enjoyable–unenjoyable, boring–interesting, pleasant–unpleasant, stressful–relaxing) components of attitude using adjectives that are commonly employed in the exercise domain (e.g., [55]. All the items are rated on 5-point bipolar adjective scale. The statement that preceded the adjectives was, “For me to participate in regular physical exercise is...”

### 3.3. Interpersonal Factors-Related Measurements

To measure social support for PA, the scale used was an adapted scale from a student survey that Amherst Health and Activity Study [56] used and validated in the previous studies with some modifications [57,58]. Initially, the scale consisted of two components: family and friends only. Researchers devised the item for teachers’ support by using the words from friends’ and family’s support [57]. It was rated on a five-point scale from 1 = never to 5 = every day. For this purpose, students were asked to answer questions related to how often they get social support from friends, family, and teachers. The item for friends/ peer support is “During a typical week at school, how often do your friends do PA or play sports with you?”. The item for teachers’ support is “During a typical week at school, how often does your teacher encourage you to do PA during recess or lunch breaks?”. The item for parents’ support is “During a week, how often does your family encourage you to do PA in your free time?”

### 3.4. Organizational Factors-Related Measurements

Organizational factors: Organizational factors were examined in this study to see the cognition of PA environment, PA policy, and PA culture in the school. The school PA environment contains equipment, facility, and safety component. The school environmental characteristics were assessed subjectively, using 10 items from the Questionnaire Assessing School PA Environment (Q-SPACE), which Robertson–Wilson and his colleagues [59] validated in previous studies in European and Asian cultures [60,61,62]. The questionnaire is composed of three factors that are rated on 5-point Likert scale from 1 = strongly disagree to 5 = strongly agree.

(1)Equipment (3 items), examining the accessibility or usability of physical equipment (e.g., there is enough equipment for PA at school);(2)Facility (4 items), measuring the accessibility or usability of PA facilities (e.g., the school ground is wide enough for PA);(3)Safety (3 items), investigating perceived safety of PA equipment and facilities (e.g., it is safe to engage in PA on the grounds and in the gym at school).

The items for measuring school PA policy and PA culture refer to a scale from a Chinese study by [41]. The items were translated into English, which is the official and institutional language in Pakistan. Translation and back-translation method were used [63]. Then, two native speakers read the instrument for confirmation of accurate translation. The instrument measures the cognition of school policy and culture for PA. The example item is, “The school organizes extra-curricular exercises during breaks”. A 5-point scale is used, ranging from 1 (strongly disagree) to 5 (strongly agree). The instrument was revalidated in the Pakistani setting, generating two factors: PA policy (7 items) and PA culture (2 items).

### 3.5. PA Measurements

Global School-Based Student Health Survey. There are many methods for PA measurement. The scale survey is the most economical among many measurement methods and is suitable for the measurement of large samples. Therefore, in this study, global school-based student health survey (GSHS) was used, which is specifically designed for measuring adolescents’ PA levels [64]. This questionnaire is already validated in Pakistan [64] and other Asian countries like China [65]. It includes four questions inquiring PA levels of the adolescents. The first question is “During the past 7 days, on how many days were you physically active for a total of at least 60 min per day?”. The responses ranged from “0 days” to “7 days.” Adolescents who were doing at least 1 h of PA per day were considered to be physically active [65,66]. The second question was “During the last 7 days, on how many days did you walk or ride a bicycle to or from school?”. Adolescents who walked or rode a bicycle to or from school for at least 3 days per week were considered to be active [65]. Third question was “During this school year, on how many days did you go to a physical education class each week?”. The responses ranged from “0 days” to “5 or more days.” Adolescents attended physical education class at least 5 days a week were classified as physically active [65]. The last question was about sedentary behavior: “How much time do you spend during a typical or usual day sitting and watching television, playing computer games, talking with friends, or doing other seated activities, such as surfing the Internet?”. The responses were “less than 1 h per day”, “1 to 2 h per day”, “3 to 4 h per day”, “5 to 6 h per day”, “7 to 8 h per day”, or “more than 8 h per day.” Adolescents who spent 3 or more hours sitting per day were considered inactive 65. In this study, the responses on only the first questions were analyzed to measure physical activity levels of adolescents as it was done in the previous study [65,66].

### 3.6. Analysis Techniques and Validation Process

IBM SPSS v.20 was used for the analyses. First, data were examined for missing values and outliers and screened accordingly. Then normal distribution was checked. Data were normally distributed, as the skewness and kurtosis were according to the acceptable standard +2 [67]. Common method bias was also tested by using common latent factor method showing no common method bias in data. Then, exploratory factor analysis was performed to ensure the underlying factor structure of all variables or resources and to check validity of instruments. Since the study was based on already developed and validated questionnaires with a determined number of factors, the number of factors was fixed to certain number as identified by α extraction. The criteria for retaining the factors were based on the eigenvalue being ≥1.0 and a value of ≥0.50 on factor loading. The 9 factors were generated under three categories. The first category included three factors: self-efficacy (9 items, α = 0.89), motivation (6 items, α = 0.85), and attitude (8 items, α = 0.87), representing individual level elements. The next factor represented interpersonal elements (3 items, α = 0.81), containing family support (1 item), friends/peer support (1 item), and teachers’ support (1 item). However, the three items were taken as separate social support indicators. The last set of 5 factors denoted organizational elements containing school PA environment, such as equipment (3 items, α = 0.74), facility (4 items, α = 0.87), and safety (3 items, α = 0.78) and PA policy (7 items, α = 0.84) and culture (2 items, α = 0.78). Under these criteria, one item was removed for culture factor. Composite reliability (CR) and average variance extracted (AVE) were calculated to check convergent validity of the instruments. All questionnaires have convergent validity possessing CR > 0.70 (CR_Equipment_ = 0.75 to CR_SE_ = 0.90) and AVE above 0.50 (AVE_attitude_ = 0.50 to AVE_culture_ = 0.64). Discriminant validity was also assessed in terms of AVE > maximum shared variance (MSV) [68]. AVE for all factors was found greater than MSV, hence ascertaining discriminant validity. This, collectively, indicates that the scale has an adequate validity in terms of convergent and discriminant validity. The results are presented in Table 1.

Correlation coefficients were computed to verify the possible relationships between PA and set of variables. At the end, stepwise forward regression in SPSS ver. 20 was applied to see the hypothesized effect of individual, interpersonal, and organizational factors on PA after controlling for demographic factors and to see the stronger predictors of PA. GC and SES were entered into the control variables. Gender, age, and BMI were excluded, as these were not found significantly related with PA.

## 4. Results

### 4.1. Demographics

Demographic characteristics of the sample is given in Table 2.

### 4.2. Preliminary Analyses

In preliminary analyses, first, mean values, and standard deviations (SD) for each variable were computed. Second, an independent sample *t*-test was applied to see the difference of gender and GC for adolescents PA. The mean PA of urban adolescents (3.24) was a little higher than that of rural adolescents (3.10). It is inferred that there was no significant difference between male and female adolescents in doing PA (*t* = −1.53, *p* > 0.05). Moreover, a significant difference was found for PA of adolescents from rural and urban areas (*t* = −2.49, *p* < 0.05). Third, one way ANOVA was applied to see the difference on PA of the adolescents from different SES. We have found students with different socioeconomic statuses were significantly different in their PA (*F* = 12.64; *p* < 0.001). Hence, both SES and GC were controlled for the major regression analyses.

Then, correlation among the variables was calculated to see the relationship of individual, interpersonal, and organizational factors with PA. Table 3 shows the relationship between PA, age, BMI, self-efficacy (SE), motivation (MOTT), attitude (ATT), family, friends’, teachers’ support, equipment, facility, safety, policy, and culture and also means and SD values for all variables. The association between PA and individual correlates depicts that age and BMI were not related to PA of adolescents. Conversely, the correlations between PA, self-efficacy, motivation, and attitude were statistically significant (r_SE_ = 0.26, r_MOT_ = 0.24, r_ATT_ = 0.38). Further, support for PA by family, friends, and teachers was significantly related to PA of adolescents (r_family_ = 0.35, r_friends_ = 0.10, r_teachers_ = 0.10). However, the coefficient for family support is quite higher than those of friends and teachers. Finally, the association among PA and organizational variables was computed, highlighting that PA is significantly related to all organizational variables, such as equipment, facility, safety, policy, and culture (r_equipment_ = 0.12, r_facility_ = 0.17, r_safety_ = 0.12, r_policy_ = 0.10, r_culture_ = 0.17). The results may be affected by the stream of COVID-19.

### 4.3. Regression Analysis

A stepwise forward regression analysis was computed to identify the main predictors of PA in a possible prediction model. In the present study, ATT, family support, self-efficacy, SES, MOT, facility, and GC factors were introduced as possible predictor variables for adolescents’ PA. R^2^ and adjusted R^2^ were calculated. In accordance with Ellis, as reported in a previous study [69], the adjusted R^2^ was reported because it represents the effect-size index that resulted after the statistical correction considering the number of participants and the variables in the prediction model.

Results of the final prediction model from the regression analysis are reported in Table 4. *R^2^* was significant (adj*R*^2^ = 0.29; *F* = 36.16, *p* < 0.001) and explained approximately 29% of the variability observed in PA. The final model included seven predictors of PA, which are attitude, family support, self-efficacy, SES, motivation, facility, and GC. The first variable that appeared as the strongest predictor in the model was attitude towards PA (*β* = 0.29, *p* < 0.001). Family support for PA was emerged as a second variable in the regression model (*β* = 0.20, *p* < 0.001). The third predictor was self-efficacy for doing PA (*β* = 0.18, *p* < 0.001). Then, SES was found as the fourth possible variable that predicted PA (*β* = 0.18, *p* < 0.001), indicating that adolescents with higher SES do more PA and vice versa. Next, motivation also played a significant role in predicting PA (*β* = 0.10, *p* < 0.05), showing that greater motivation would lead to higher PA. Additionally, facility for PA predicted PA of adolescents (*β* = 0.09, *p <* 0.05), presenting that the more facilities for physical exercise, the higher PA they reported. GC also played a role in predicting PA (*β* = 0.10, *p <* 0.05). Other variables, such as peers’ support, teachers’ support, equipment, and safety, policy, and PA culture, did not emerge as predictors of PA in the present stepwise regression analysis. Although some influences of these factors on PA could be seen from previous correlation analyses (as reported in Table 3), they were not recognized as significant predictors of PA in the regression model.

## 5. Discussion

In the current study, within the social ecological framework, associations between individual, social, and organizational factors and participation in PA among Pakistani adolescents were examined. In addition, the variables predicting PA were identified. Prior to the examination of the theoretical relationships among the study variables, we validated all the instruments in the Pakistani setting that allowed us to subsequently test the hypothesized associations.

Our results provide evidence of significant correlations between all the individual, interpersonal, and organizational level variables. The regression analysis showed that attitude, family support, self-efficacy, SES, motivation, facility, and GC influenced PA. Results from regression models provided support for our hypotheses that various theoretical factors underlying intrapersonal or individual-level resources are associated with Pakistani adolescents’ participation in PA. However, the hypothesis that interpersonal resources would affect PA was partly accepted, as only family support emerged as a predictor of PA, while peers’ and teachers’ support did not. With respect to factors related to individual resources, the results show that adequate self-efficacy regarding participation in PA, high levels of motivation, attitude towards PA, socioeconomic status, and GC were more likely to be associated with the PA level of Pakistani adolescents. However, gender, age, and BMI were not related to PA, suggesting that adolescents participate in PA irrespective of their gender, age, and BMI. These findings are consistent with other reports that self-efficacy, motivation, and attitudes are important in initiating and maintaining PA [15,70].

On the influence of socioeconomic and GC, our findings suggest that both socioeconomic status and GC were associated with participation in PA. These findings are consistent with a previous study indicating that adolescents having middle or low socioeconomic status tended to exercise less compared to those having higher socioeconomic status [27,29]. Our results also authenticate the findings of previous research [71], who compared adolescents (11–16 years old) from different socio-economic backgrounds in Pakistani schools and found a significant difference in their PA.

With regard to GC, past research exhibited that adolescents from rural areas were more active than those from urban areas [72]. The present research is opposed to these findings. Our findings also contradict another survey study in Pakistan that indicates that 67% of the rural Pakistani adolescents were active as compared to 35% in urban areas [73]. Conversely, the current results are similar to a recent systematic review and meta-analysis in Pakistan that provided evidence that adolescents from rural areas had lower PA than those in urban areas [74]. Our findings suggest that there was a significant effect of support for PA from family on PA participation by the adolescents and are in line with the view that social support for PA from family is likely to facilitate PA participation [7]. Moreover, no significant effect was found for friends’/peers’ and teachers’ support. This result is opposite to the previous research that social support for PA from friends/peers and teachers lead to increase in adolescents’ PA [7,15,75,76]. However, there are a few examples where teachers’ and peers’ support do not predict PA level [62,77]. Although not specifically measured in the current study, it is likely that social support may affect PA indirectly through one’s perception of self-efficacy [78], which has been shown to consistently predict PA [78,79]. Hence, future studies could be conducted to see the mediating effect of self-efficacy between social support and adolescents’ PA. Moreover, adolescents getting social support from family were shown to correlate with PA. This is consistent with a systematic review conducted in Thailand that demonstrated that adolescents who participated in PA with their parents were likely to report higher levels of leisure time PA [15].

Similar to the previous work [45,46], organizational factors exhibited inconsistent results. Among organizational factors, only PA facility was shown to be important with regard to PA participation among school adolescents in Pakistan. Our findings suggest that factors such as school PA-related facilities were likely to be associated with increased levels of PA among school adolescents. It is in line with the previous studies conducted in other Asian countries like Japan, which reported that positive social and built environmental features, such as availability and accessibility to PA facilities, were associated with increased PA [61]. Our study further elaborates that school PA equipment, safety, policy, and culture did not have any effect on PA participation in accordance with the past research [61]. However, the study results were consistent with the past research exhibiting no consistent or significant evidence for environmental and policy correlates of PA [15]. Further research is needed in to verify the environmental and policy correlates of PA. Although not specifically measured in the current study, it is likely that equipment, safety and PA policy, and PA culture may affect PA indirectly through self-efficacy or social support [61], which has been shown to consistently predict PA [79]. Hence, future studies could be conducted to see the mediating effect of self-efficacy and social support between environmental factors and adolescents’ PA. Moreover, the results may be affected by COVID-19 in the country. The situation may be different under normal circumstances. Further investigations are thus required for better results, providing basis for future interventions.

### Strengths, Limitations, and Future Research Directions

This study has several strengths. First, it is one of the few studies that specifically evaluated the influences of individual, social, and organizational factors on PA in the context of Pakistani adolescents. Next, the study provided a scientific validation of the social ecological model in the context of PA participation in Pakistan, where it has not been tested before. However, the study was cross-sectional in nature and is conducted in only one city of Pakistan, which might affect the generalizability of the results. Further, the cross-sectional design may limit the causal inferences. Future intervention and longitudinal studies are required to better understand the effect of various factors affecting adolescents’ PA participation over time.

To improve adolescents’ health, health promotion initiatives should be intended to highlight the benefits associated with being physically active and the importance of adolescents’ perceptions of individual, interpersonal, and organizational level factors. As increasing exercise self-efficacy, exercise motivation, and exercise attitude encourages the individuals to participate in PA, interventions considering these concepts would be beneficial in future. Further, parental, peer, and teacher support for PA could provide better opportunities for adolescents to be active and have the benefits of more significant PA behavior. 

It would be effective to investigate in future studies the facilitators and barriers that parents, teachers, and peers encounter regarding providing support to be physically active. In addition, the study also demonstrated the significance of the school environment in promoting adolescents’ PA. However, the organizational factors did not predict PA except for PA facility. The reason may be that the study was conducted during the COVID-19 pandemic in the country. Hence, we call for future researchers to investigate these factors in normal situations. Future research should also focus on identifying specific elements of school environments and devising interventions that encourage higher levels of PA. Furthermore, future efforts should focus on mechanisms through which some of the multilevel factors may either mediate or moderate PA outcomes. For example, a study by Ren and his colleagues [78] suggested that individual perceptions of self-efficacy play an important role in mediating the relationships between social support and PA participation by the adolescents; it showed that social support influences PA indirectly through one’s intrinsic and extrinsic motivation. In addition, social and physical environments may interact with individual-level factors. For example, school environmental factors may be important in facilitating PA in adolescents; however, individual factors, such as self-efficacy and motivation, may influence people in using these available resources to engage in PA. Finally, given that most PA takes place in social and built environments [61], a multilevel approach is needed to enhance PA of adolescents. This approach will allow us to identify potentially modifiable factors that can inform policies and facilitate the design of interventions to change PA behavior of the individuals.

## 6. Conclusions

The present study provided validation of the social ecological model explaining PA among Pakistani adolescents. The purpose of this study was to see the effect of individual, social, and organizational factors on PA of adolescents in Pakistani schools. It is found that self-efficacy, motivation, and attitude significantly predicted the PA of Pakistani adolescents. Conversely, only family support was associated with PA participation of adolescents, but teachers’ and peers’ support were not. Further, PA facility among organizational variables had significant relationship with PA, while equipment, safety, PA policy, and PA culture did not have any association with PA. The results offer an insight into the potential role of multilevel factors for promoting PA, suggesting that public health intervention strategies for promoting PA in adolescents should recognize multiple levels of influences that may either enhance or impede the likelihood of PA among adolescents.

## Figures and Tables

**Table 1 ijerph-18-07011-t001:** Validation of Tools.

Variables and Items	Items and Estimates									AVE	√AVE	C.R.
Self-efficacy (se)	se1	se2	se3	se4	se5	se6	se7	se8	se9	0.51	0.71	0.90
Estimate	0.672	0.716	0.789	0.642	0.756	0.798	0.675	0.692	0.65			
Motivation (MOT)	mot1	mot2	mot3	mot4	mot5	mot6				0.52	0.72	0.86
Estimate	0.682	0.791	0.743	0.638	0.693	0.759						
Attitude (att)	att1	att2	att3	att4	att5	att6	att7	att8		0.50	0.71	0.89
Estimate	0.697	0.71	0.732	0.673	0.792	0.691	0.764	0.597				
Social support (ss)	ss1	ss2	ss3							0.61	0.78	0.82
Estimate	0.852	0.684	0.795									
Equipment	equip1	equip2	equip3							0.51	0.72	0.76
Estimate	0.632	0.810	0.697									
Facility	fac1	fac2	fac3	fac4						0.62	0.78	0.86
Estimate	0.835	0.694	0.763	0.821								
Safety	saf1	saf2	saf3							0.56	0.75	0.79
Estimate	0.761	0.687	0.793									
Policy	pol1	pol2	pol3	pol4	pol5	pol6	pol7			0.55	0.74	0.89
Estimate	0.823	0.685	0.746	0.682	0.842	0.724	0.674					
Culture	cul1	cul2								0.64	0.80	0.78
Estimate	0.795	0.813										

**Table 2 ijerph-18-07011-t002:** Demographic characteristics of the participants.

Demographic Variable (*n* = 618)	Frequency	Percentage
School		
Girls	2	50%
Boys	2	50%
Class		
6th grade	217	35.1
7th grade	206	33.3
8th grade	195	31.6
Gender		
male	318	51.5
female	300	48.5
Age		
11 years	94	15.2
12 years	165	26.7
13 years	234	37.9
14 years	125	20.2
SES		
Lower class	243	39.3
Middle class	366	59.2
Upper middle class	9	1.5
GC		
Urban	311	49.7
Rural	307	50.3
BMI = weight (kg)/height^2^ (m)		
Mean	17.05	
SD	2.31	

**Table 3 ijerph-18-07011-t003:** Correlation between PA and individual, interpersonal, and organizational variables.

	PA	Age	BMI	SE	MOT	ATT	Family Support	Friends’ Support	Teachers’ Support	Equipment	Facility	Safety	Policy	Culture	Mean	SD
PA	1														3.17	0.69
Age	0.071	1													12.63	0.97
BMI	0.046	0.210 **	1												17.05	2.31
SE	0.261 **	−0.055	0.037	1											2.86	0.69
MOT	0.238 **	0.131 **	0.039	0.195 **	1										2.86	0.73
ATT	0.380 **	0.039	0.068	0.120 **	0.221 **	1									2.77	0.59
Family Support	0.345 **	−0.006	−0.005	0.089 *	0.130 **	0.262 **	1								3.27	0.80
Friends’ Support	0.100 *	0.021	0.073	0.009	0.050	0.171 **	0.032	1							3.06	1.19
Teachers’ Support	0.101 *	0.023	0.074	0.003	0.050	0.166 **	0.036	0.995 **	1						3.06	1.19
Equipment	0.124 **	−0.046	0.103 *	0.074	0.098 *	0.225 **	0.043	0.470 **	0.472 **	1					2.68	0.86
Facility	0.167 **	−0.054	0.100 *	0.049	0.071	0.206 **	0.039	0.404 **	0.410 **	0.705 **	1				2.68	0.76
Safety	0.118 **	−0.016	0.093 *	0.087 *	0.043	0.125 **	0.035	0.324 **	0.327 **	0.458 **	0.655 **	1			2.74	0.80
Policy	0.087 *	−0.042	0.082 *	0.013	0.005	0.161 **	0.001	0.359 **	0.359 **	0.534 **	0.552 **	0.394 **	1		2.88	0.80
Culture	0.173 **	−0.040	0.137 **	0.009	0.018	0.256 **	0.194 **	0.312 **	0.312 **	0.368 **	0.337 **	0.203 **	0.356 **	1	2.98	0.66

* Correlation is significant at the 0.05 level (2-tailed). ** Correlation is significant at the 0.01 level (2-tailed). PA, physical activity; BMI, body mass index; SE, self-efficacy; MOT, motivation; ATT, attitude.

**Table 4 ijerph-18-07011-t004:** Final model of the stepwise regression analysis for predicting PA.

Predictors	adj*R*^2^	*Β*	*t*	*F*	*p*
1. Attitude	0.143	0.291	6.733	104.207	0.000
2. Family Support	0.207	0.198	6.334	81.501	0.000
3. SE	0.247	0.183	5.193	68.382	0.000
4. SES	0.264	0.177	3.769	56.279	0.000
5. Motivation	0.274	0.102	2.995	47.507	0.003
6. Facility	0.280	0.086	2.674	41.087	0.008
7. GC	0.285	0.106	2.232	36.159	0.026

## Data Availability

The data presented in this study are available on request from the corresponding author. The data are not publicly available due to research ethics.

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
