# Peer review of "Individual, Interpersonal, and Organizational Factors Affecting Physical Activity of School Adolescents in Pakistan"

_ijerph, 2021, doi:10.3390/ijerph18137011_

Round 1

Reviewer 1 Report

Apologies for the delay in getting this review to you. I would like to commend the authors on conducting a project of this scale, during restrictions that would likely otherwise have impacted research due to the pandemic. 

The work is very well referenced, perhaps overly so, given the nature of the project. This is best reflected in the introduction which whilst thorough, provides a large amount of background information. This would fit well within an MSc of PhD but for a paper, I feel the introduction needs to be more direct and lead the researcher to the research question more directly. Put more simply, the paper at present is unbalanced, with such a large introduction compared to a short discussion.

Methods - your participants and procedure section appears to be written in a manner that better reflects a results section. Can this be condensed or presented in a table?

The measures section, specifically the intrapersonal factors related measurements, contains a lot of background and critical points that may better be used in the discussion, or in a condensed introduction. There is also a mixture of tenses within the writing, which needs revision, and affects the readability of the paper.

The analysis and validation process is very thorough. You seemed to have performed more analyses than are reported within your results section though? Again, a bit like the earlier methods sections, this paragraph reports a mixture of procedure and results. This needs revision. Magnitude of correlations require interpretation here, as in the results all appear to be small despite significance.

Results - line 374 please check this p value. It can't be <0.000, perhaps <0.001?

Given that your correlations are all 'small', albeit statistically significant, is a hierarchical regression model appropriate?  Would a forced regression not permit a simpler, potentially less biased investigation of all variables simultaneously? Or a stepwise forward model? You are relying on a lot of small correlations to build a model that predicts complex behaviour.

I'm not clear which is your preferred regression model? Model 2 appears to be a balance between significant variables, improvement in R2 and number of variables within the model? On a related note, how do equipment and facility differ? Likewise, how do culture and support differ? This is somewhat important given the progression of your models, if you choose not to rerun a different type of regression.

Discussion - this is very short  given the depth of your analyses, and the potential for discussion on such a complex topic, as outlined in your introduction. Redress this balance please.

439 - 431 - how does this differ to your findings? How is the review so different from the Malik findings?

450 - 451 - could you run an intraclass correlation coefficient or similar reliability statistic between the data obtained for social support and self-efficacy? Your current correlations suggest this may not support previous findings with correlations reported <0.10

Your future research paragraph needs some work and refinement but is possibly the most critical paragraph in the paper. We need more of this type of thinking throughout, please.

Reference formatting throughout requires revision to align with journal guidelines.

Reviewer 2 Report

Reviewer report

General comment

The study “Individual, Interpersonal, and organizational factors affecting  physical activity of school adolescents” was as proposal to explore individual, interpersonal and organizational factors that may influence the physical activity (PA) of adolescents (ages 10–14) in Pakistani schools.

Specific comments

Line 42: Please add a newer reference about meeting recommended guidelines for physical activity.

Line 54: I have doubts as to whether the geographical characteristics correspond to individual characteristics. Please explain or justify better.

Line 56: Please review the spaces between words. Same review in all text.

Line 60: Please replace physical activity by PA. Change this throughout the text. “PA” was used in abstract. You can reuse it in the intro.

Line 74: Please provide cites from adolescents’ studies. For example, the reference of Lizana et al. incorporates children. Review all your references and replace them.

Line 85: What about junior high school? Please use the same concept throughout the text. This will help to understand better.

Line 93-96: The sentence is very long, please separate to explain better the idea.

Lines 110-113: Please add respective references about Parents’ PA.

Line 114: The reference of Célis-Morales was made in 6 years old children. Not is possible to use as reference in this context, when the group of the study are adolescents.

Lines 121-139: Be careful with claims about parental support for children, which is correct. But the same is not the case with adolescents, where the association of support is lost.

The main problem is that it uses child references for a study done in adolescents and that is not the most correct.

Methods.

In abstract was explained that the students were randomized, but this important point has not been explained in this section.

Line 226-228: Use the verbs in simple past, not in future.

Review all section and delete “will be” and use “was” or “were”.

Lines 289: Please add “comma” in the sentence in bold.

Lines 293, 298 and 311: Please change “tool” by “questionnaire” or “instrument”

Results.

Please add a sociodemographic table and explain the results about it.

Discussion.

Line 436: Please replace physical activity by PA-.

Lines 448-453: Explain in a better way, because there is no relationship between physical activity and support from peers and teachers unlike other studies.

What you have said about it does not have a scientific weight.

Line 453: why have they not done mediation in this study?

Line 480: That study was based on the social ecological model, is not a strength.

Line 482: it is very pretentious to write “The study has a few limitations as well”, since the limitations described are very important. I recommend delete the sentence.

Conclusion.

The conclusion is very general and should mention the main factors that affect the PA of the adolescents.

References:

In Celis-Morales, the information is missing

Review the Yi, J. (2014) and complete the missing information.

Final comments.

The manuscript is interesting above all as scientific information for the country of origin.

However, I detect two big problems.

  1. The article is very long, especially the introduction and the content has not been summarized so that the reader has a clearer reading. This could affect its possible citation if it is published.
  2. The references have not been used correctly. Several correspond to studies with children that cannot be used in this study, because the behavior and factors associated with AP are very different. Also, several other citations in References are misspelled.

I suggested considerably reducing the introduction and methodology to make it easier to understand.

Reviewer 3 Report

This cross-sectional study showed that factors at multiple levels based on ecological models were associated with physical activity (PA) in Pakistani adolescents. Since there are many studies using ecological models to predict adolescent PA, the topic is not novel. The authors underscore that it is necessary to study how different cultures may have a role in PA. However, the authors did not provide any hypotheses what the differences in cultures between Pakistan and other countries were and how the differences might affect PA. The authors also did not follow a citation style for the journal at all. There are other concerns as follows.

  1. Although the survey was conducted during the COVID-19 pandemic, the authors did not mention the potential influence of the new normal on the survey and findings.
  2. The authors should elaborate on the sampling method. How many students were as the study population? How were the students selected? How many and what types of schools were participated in the study and how were the schools selected? Is this cluster sampling? Were the questionnaire surveys conducted in a classroom, home, or other places? What is a selection bias?
  3. How were urban/rural areas defined? How were lower/middle/upper social classes defined?
  4. Why did the authors choose each measurement related to three-level factors, such as intrapersonal, interpersonal, and organizational factors? The authors should refer to the rationale for choosing those measurements.
  5. In this study, PA levels were measured by 3 questions in the Global School-based Student Health Survey. However, only one PA variable appeared in the Results section. The authors should address this point.
  6. On line 342, the authors mentioned that “the 9 factors were generated under three categories”. How did the authors get the finding? The authors should test the fit of the higher-order factor analysis model, which consists of the 9 factors as the first-order factors and the 3 categories as the second-order factors.

Round 2

Reviewer 1 Report

Thank you for a revised version of the manuscript. It is clear that extensive comments and revisions have been provided in response to reviewers' comments.

Thank you for a clearer presentation of the regression model.

I am still struggling to find the reliability results that you mention in your response to me, however? You report them as being done, and I think your methods reflect this, but the results are not clearly stated. Please note I am not concerned with alpha values here, but that the r values presented in your correlations are very low. Whilst they may be statistically significant, this does not mean that they are not weak in magnitude.

Reviewer 2 Report

Thanks to authors for review and correct the manuscript which improved a lot.

Only things of format need to review. In edition process can change.

Reviewer 3 Report

Overall, the paper has not sufficiently addressed the reviewer’s comments.

The authors did not respond to what the differences in cultures between Pakistan and other countries were and how the differences might affect PA.

2. The authors did not respond to whether the questionnaire surveys were conducted in a classroom, home, or other places and what a selection bias is. Additionally, there is a contradictory description, that is, the abstract described “students randomly selected”, but the text mentioned, “the sample was conveniently taken”.

3. The authors did not respond to this comment. The definitions of areas and social classes are unclear.

5. The reason why only one PA question was analyzed, not other GSHS items, was not clear.

6. The authors did not respond to how the authors got the finding that “the 9 factors were generated under three categories”. The authors should test the fit of the higher-order factor analysis model, which consists of the 9 factors as the first-order factors and the 3 categories as the second-order factors. The authors should demonstrate the findings of factor analyses.
